



# Relating North Atlantic Deep Water transport to ocean bottom pressure variations as a target for satellite gravimetry missions

Linus Shihora[1], Torge Martin[2], Anna Christina Hans[2], Rebecca Hummels[2], Michael Schindelegger[3], and Henryk Dobslaw[1]

[1]GFZ German Research Centre for Geosciences, Department 1: Geodesy, Potsdam, Germany
[2]GEOMAR Helmholtz Centre for Ocean Research Kiel, Kiel, Germany
[3]Institute of Geodesy and Geoinformation, University of Bonn, Bonn, Germany

**Correspondence:** Linus Shihora (linus.shihora@gfz-potsdam.de)

**Abstract.** The Atlantic Meridional Overturning Circulation (AMOC) is a salient feature of the climate system, observed for its strength and variability with a wide range of offshore installations and expensive sea-going expeditions. Satellite-based measurements of mass changes in the Earth system, such as from the Gravity Recovery and Climate Experiment (GRACE) mission, may help monitor these transport variations at large scale, by measuring associated changes in ocean bottom pressure
(OBP) at the boundaries of the Atlantic remotely from space. However, as these signals are mainly confined to the continental slope and small in magnitude, their detection using gravimentry will likely require specialised approaches. Here we use the output of a fine-resolution (1/20°) regional ocean model to assess the connection between OBP signals at the western boundary of the North and South Atlantic. We find that North Atlantic Deep Water (NADW) transports in the ∼1–3-km depth range can be reconstructed using spatially averaged OBP signals with correlations of 0.75 (0.72) for the North (South) Atlantic and root-
mean-square errors of ∼1 Sverdrup on monthly to interannual time scales. We further create a synthetic dataset containing only OBP signals due to NADW transport anomalies at the western boundary, which can be included in dedicated satellite gravimetry simulations to assess the AMOC detection capabilities of future mission scenarios and to develop specialised recovery strategies that are needed to track those weak signatures in the time-variable gravity field.

## 1 Introduction

The Atlantic Meridional Overturning Circulation (AMOC) is a defining element of the three-dimensional ocean general circulation and comprises of a fine-structured system of currents. In aggregate, the upper overturning cell transports warm water masses in the near-surface layers of the Atlantic northwards. In the Labrador Sea and Nordic Seas, comparatively dense water masses are formed by deep convection and overflows, respectively. The resulting deep water mass, the so called North Atlantic Deep Water (NADW), is subsequently transported southward for reasons of continuity and rises in the Southern Ocean, e.g.,
in Antarctic divergence Buckley and Marshall (2016). Additionally, there is a deep overturning cell which contributed to the AMOC and is associated with the transport of Antarctic Bottom Water (AABW). AABW is formed in the Southern Ocean close to Antarctica in, e.g., the Weddel and Ross seas where extremely cold and salty water masses are produced and form



the deepest water masses in the ocean. Part of the AABW then flows northward along the ocean floor initially west of the mid-Atlantic ridge (Lozier, 2010; McCarthy et al., 2020).

Whereas the detailed influence of the AMOC in climate change scenarios is still debated, evidence suggests that the AMOC has not only a regional impact on the North Atlantic but also global consequences (Collins et al., 2022). Over the past decades, this motivated numerous sea-going campaigns and long-term observation arrays to measure variations in meridional transports at various latitudes in the North and South Atlantic (Frajka-Williams et al., 2019; McCarthy et al., 2020). While these in-situ measurements are certainly the most direct approach to monitor the AMOC, they come with significant annual operations costs

so that measurement arrays are still relatively sparse.

    Instead of basin-wide measurement arrays, zonally integrated transport variations could also be inferred through the measurement of ocean bottom pressure (OBP) variations at the sloped lateral boundaries of the Atlantic basin alone. Under the geostrophic approximation, meridional transport variations can be determined through the difference between eastern and western boundary pressure. In fact, as shown by Bingham and Hughes (2008), the western pressure signals alone are al-

ready sufficient to capture almost all of the geostrophic transport variations at 42°N on inter-annual time-scales. This physical connection offers the prospect of determining AMOC variations through satellite gravimetry measurements such as GRACE (Tapley et al., 2004) and GRACE-FO (Landerer et al., 2020), which measure large-scale OBP anomalies globally with monthly resolution.

    Some success with this approach has indeed been reported by Landerer et al. (2015), who used JPL RL05 GRACE mascon

solutions to determine Lower North Atlantic Deep Water (LNADW, depths of 3–5 km) transports at 26.5°N over the period 2003–2014. The authors found good agreement with in-situ measurements from the RAPID array at the very same latitude. These results have been met with scepticism by Hughes et al. (2018), who suggest that the transports inferred by Landerer et al. (2015) are not necessarily associated with the western continental slope dynamics. While it may be that the smoothing effect of GRACE measurements allows for the detection of transport-related, spatially coherent OBP signals, these OBP variations

have been shown to be confined to the narrow continental slope only (Bingham and Hughes, 2008; Roussenov et al., 2008; McCarthy et al., 2020).

    There are, however, two considerations which motivate further studies of GRACE-based AMOC variability. First, all attempts so far rely on 'standard' global monthly gravity field solutions, which are in no way tuned for the detection of the narrow OBP signals along the continental slope. Specific tailored approaches, such as gravity field inversions with averaging

times other than monthly or spatial constraints emphasizing the continental slope, may lead to better signal-to-noise ratios for the OBP signals caused by AMOC variations. Secondly, space agencies around the world are currently preparing new satellite gravimetry missions such as GRACE-C (by NASA and DLR with an expected launch in 2028) and NGGM (planned by ESA for 2032), which together will form the MAGIC constellation. MAGIC offers the prospect of resolving much smaller spatial scales than previously accessible from GRACE alone (Pail et al., 2015). Preliminary analyses of the capabilities of the MAGIC

constellation show that the increases in resolution may be sufficient to capture transport related boundary pressure signals in the northern Atlantic (Daras et al., 2023). The MAGIC Mission Requirements Document indicates a target resolution and accuracy



of 100 km and 1.5 cm equivalent water height at monthly time-scales (Haagmans and Tsaoussi, 2020), but detailed end-to-end simulations with the final mission configuration are still to be performed in the near future.

We therefore propose to pursue a more thorough assessment of the capabilities of current and future satellite gravimetry missions to monitor AMOC transport variability. In the present work, we analyse the connection between NADW transport variations and OBP in a high-resolution general circulation model that is covering both the North and South Atlantic. We explore how the modelled OBP variations associated with geostrophic meridional transports can be synthesised in preparation of dedicated end-to-end simulation studies of satellite gravimetry. This in turn will allow future studies to design suitable geodetic processing strategies (considering, e.g., various noise contributors including ocean tides) or assess planned future mission scenarios such as MAGIC. We limit our analyses to periods below 5 years, corresponding to the nominal mission lifetime of gravity missions in low-altitude orbits.

## 2 Methods

### 2.1 Inferring Meridional Transport Variations from Ocean Bottom Pressure

Meridional transports can be related to boundary pressure based on the zonal momentum equation under geostrophic approximation (Roussenov et al., 2008)

$$fv = \frac{1}{\rho_0}\frac{\partial p}{\partial x}, \tag{1}$$

where $v$ represents the meridional geostrophic velocity, $p$ the pressure, $f = 2\Omega\sin\phi$ the latitude-dependent Coriolis parameter that is induced by Earth's rotational frequency $\Omega$, and $\rho_0$ the reference density. Integrating in the zonal ($x$) direction over the entire Atlantic basin gives the meridional geostrophic transport $T$.

$$T(y,z) = \int v\,\mathrm{d}x = \frac{p_E(y,z) - p_W(y,z)}{f\rho_0} \tag{2}$$

where $p_E$ ($p_W$) is the eastern (western) boundary pressure. As highlighted by Bingham and Hughes (2008), the transport variations are already represented quite well in the boundary pressure at the western boundary alone. As a result, measurements of only western boundary pressure from in-situ recorders (or satellite gravimetry) may be used to infer anomalies of meridional transports via

$$T'(y,z) \approx -\frac{p'_W(y,z)}{f\rho_0}. \tag{3}$$

Instead of the transport at a certain depth, the total transport anomaly over a given depth range can be determined by vertical integration. For the NADW transport anomalies we focus on here this gives:

$$T'_{NADW}(y) \approx -\int\limits_{3000\ m}^{1000\ m} \frac{p'_W(y,z)}{f\rho_0}dz. \tag{4}$$



There are, of course, simplifying assumptions included above. For one, we are presuming that geostrophic balance holds for these large scale flows, especially when integrating zonally through a western boundary current. As long as the boundary current is narrow and oriented northwards, the impact of non-geostrophic terms should be minimal (Bingham and Hughes, 2008). In addition, since frictional terms are neglected in the above framework we cannot trace transport variations in the Ekman layer. Moreover, bottom friction can only be neglected when the sidewalls of the basin are sufficiently steep (Little et al., 2019).

Secondly, ignoring contributions from the eastern boundary means that the net meridional transport is inaccessible as well as, e.g., basin wide modes. Despite these limitations, several simulation studies have confirmed the tight connection between western boundary pressure and transport variations (Bingham and Hughes, 2008; Roussenov et al., 2008).

    Since we are interested in inferring variations in the upper overturning cell, there are two water masses to consider. Typically, the overturning at a given latitude is taken to be the maximum of the streamfunction at that particular latitude (McCarthy et al.,

2020) and thus comprises the northward flowing upper layer of the ocean up to a depth of about 1000 m. OBP variations associated with this upper limb of the AMOC can thus only be sensed in areas shallower than this threshold, which is the continental shelf region. Alternatively, changes in the overturning can be quantified by variations in the southward flowing limb mainly associated with the NADW transport at depths below 1000 m and extending as deep as 5000 m (Send et al., 2011). For NADW transport anomalies, western boundary pressure variations are found primarily along the continental slope which

connects shelf and deep ocean regions. In principle, both of these approaches should yield equivalent results since north- and southward mass transports of the upper overturning cell mostly compensate for the North Atlantic. In the southern Atlantic, the flow of AABW somewhat complicates the relation between northward and southward limb of the upper overturning cell which means that the NADW transport does not equal the maximum of the streamfunction anymore.

    However, an advantage of considering OBP variations along the continental slope (indicative of NADW variations) is that

the overall variability in OBP in the slope region is significantly smaller compared to adjoining deep ocean and continental shelf areas, where either eddy activity or wind-driven circulations dominate (cf. Fig. 1). We thus expect transport-induced variations to make up a larger percentage of the total variability on the continental slope. Yet this also means that the OBP signals are of a very small horizontal extent, which poses a challenge for detecting them in satellite gravimetry observations.

### 2.1.1   Ocean Model

We rely in this study on simulations with the ocean-sea ice model VIKING20X, which is based on the NEMO3.6 ocean (Madec et al., 2023) and LIM2 sea-ice models (Fichefet and Maqueda, 1997) being executed on the global eddy-permitting ORCA025 tripolar Arakawa-C grid with a nominal resolution of 0.25°. VIKKING20X features regional refinement for the Atlantic Ocean from 33.5°S to about 65°N with a nominal horizontal resolution of 1/20°applying 2-way nesting by means of Adaptive Grid Refinement in Fortran (AGRIF, Debreu et al. (2008)), which enables an eddy-rich simulation in this specified area. Both, the

global low and regional high-resolution grids consist of 46 vertical z-layers. The high horizontal resolution in the Atlantic is well suited for the investigation of the boundary pressure changes along the narrow western continental slope. The model run we employ here, VIKING20X-JRA-OMIP internally named VIKING20X.L46-KFS003 (Biastoch et al., 2021), extends from 1958 to 2019 applying JRA55-do atmospheric forcing (Tsujino et al., 2018). From the model output, we derive the Atlanitc



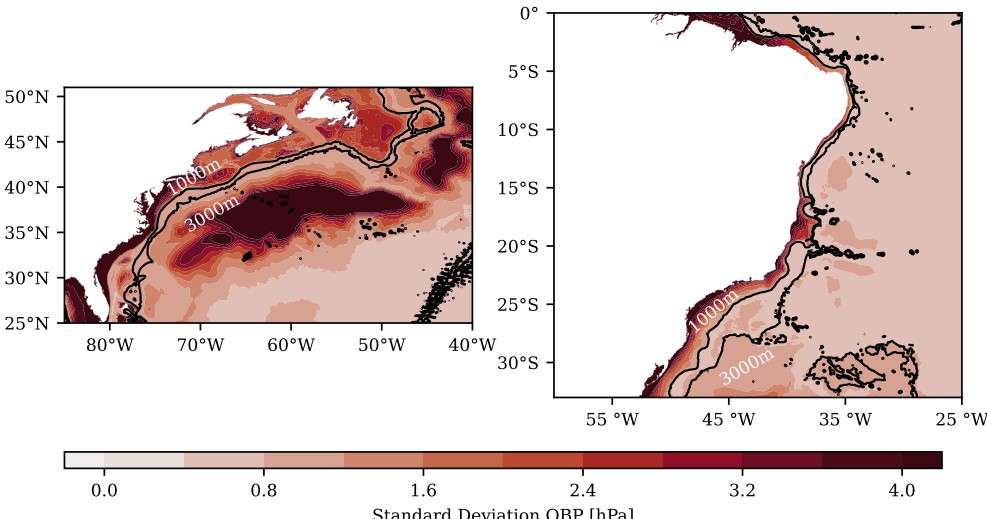

**Figure 1.** Standard deviation of ocean bottom pressure from VIKING20X for parts of the North (left) and South Atlantic (right). 1000 and 3000-m isobaths are shown as contour lines indicating the continental slope region.

meridional overturning stream function computed by zonally and vertically integrating the daily mean meridional velocity

component and ocean bottom pressure (OBP). OBP is computed using the function *cdfbotpressure -ssh2* of the CDFTOOLS package (Akuetevi et al.), i.e. taking the sea surface height output into account using the modelled 2-D surface density. We limit our analysis to the simulations years from 1970 onward to avoid potential impacts by model spin-up. Further, high-pass filtered is applied to all quantities prior to analysis with a cut-off period of 5 years in order to remove any influence by long-term model drift. For further details on the model configuration we refer the reader to Biastoch et al. (2021).

VIKING20X was previously evaluated in great detail for its representation of the AMOC (Biastoch et al., 2021), the deep convection in the Labrador Sea (Rühs et al., 2021) and the Deep Western Boundary Current (Handmann et al., 2018). Overall, the model has been found to be of compatible, high quality with regard to other ocean models (Hirschi et al., 2020). We summarize some relevant details here for proof of reliability. Validation of the simulated basin-wide circulation with in-situ measurements, such as from RAPID, OSNAP west and 11°S sections (Biastoch et al., 2021), show good agreement in terms of

the vertical structure and long-term variability. However, on inter-annual time-scales, VIKING20X slightly underestimates the variability at 26.5°N. While the upper overturning cell related to the AMOC has a fairly realistic vertical structure, the deeper overturning cell is characterised by a vertical displacement of the transition from southward flowing lower NADW to northwards directed AABW by about 500 m. In terms of the temporal variations, we note, however, that an exact representation of the oceanic state is not necessarily required for satellite simulation studies. Instead, it is vital that the spatio-temporal variations

in the overturning are realistic. The VIKING20X model run with its high spatial resolution and overall good representation of the AMOC is thus well suited for our purposes.



## 3 North Atlantic

At first, we investigate the connection between western boundary pressure anomalies and variations in the upper NADW transports in the northern Atlantic. This is done by using (i) the OBP signal at the continental slope of the western boundary at depths between 1000 m and 3000 m, and (ii) the OBP signals from both the slope and shelf, i.e., above 1000 m. Additionally, we investigate (iii) the deeper ocean region between 3000 m and 5000 m connected to the lower NADW, as examined by Landerer et al. (2015), to test whether the signals reported in that study are reproducible with VIKING20X.

We specifically focus on the region between 25°N and 40°N. At the southern margin, this includes the location of RAPID at 26.5°N. We choose a northern limit of 40°N for two reasons: For one, when considering variations in meridional transport, we need to ensure that we stay clear of the region of deep water formation in the North Atlantic. Secondly, as indicated by Bingham and Hughes (2008), ageostrophic contributions become more relevant when the zonal integral includes a significant along-stream component which is the case for the North Atlantic Current. Additionally, there is a strong eddy contribution after the current's separation from the coast. Limiting the maximum latitude to 40°N helps to minimise these complications. To illustrate the meridional connectivity in transports over the chosen interval, we show a Hovmöller diagram based on the VIKING20X model transport anomalies in the 1000–3000 m depth range in Fig. 2, where horizontal dashed lines illustrate the latitude band chosen. We find a good latitudinal coherence for both the 1–5-year band and also for periods longer than 5 years. Although here we assess the lower part of the overturning cell compared to the upper part shown in Biastoch et al. (2021), the results concerning the long-term evolution are rather similar despite the additional fact that here transport anomalies averaged in z-coordinates are used compared to sigma-coordinates which are more commonly used in the subpolar North Atlantic.

For the interval 25–40°N, we derive a single time series of monthly upper NADW transport anomalies by using the transports derived from the model streamfunction in the 1000–3000 m depth range and subtracting a mean value of 17.3 Sv. The result is shown in Fig. 3 with a 5-year high-pass filter applied (a) and a 1–5-year band-pass filter applied (b). We will, in the following, largely disregard transport variations on time-scales longer than 5 years, since satellite missions operating in the low Earth orbit of approximately 500 km usually have a nominal mission life-time of only 5 years due to atmospheric drag effects. Inferring variations at longer periods will therefore require connecting measurements from more than one satellite mission which adds additional challenges and is largely out-of-scope in future mission design considerations. Thus, we will primarily use the filtered time series of upper NADW transport variations, as shown in Fig. 3a, to investigate the connection to OBP variations along the western North Atlantic in the upcoming sections.

### 3.1 Integration Approach

We now connect the filtered model upper NADW transport anomalies to OBP signals along the western continental slope of the North Atlantic. Monthly OBP fields from VIKING20X are filtered in the same way as the NADW transports using either a 5-year high-pass filter or a 1 to 5-year band-pass filter. Next, we calculate and plot the Pearson correlation between the model transport anomaly time series and the correspondingly filtered OBP anomalies for each grid point in the region of the western continental slope in Fig. 4. Note that we have here used the negative of the transport time series which essentially





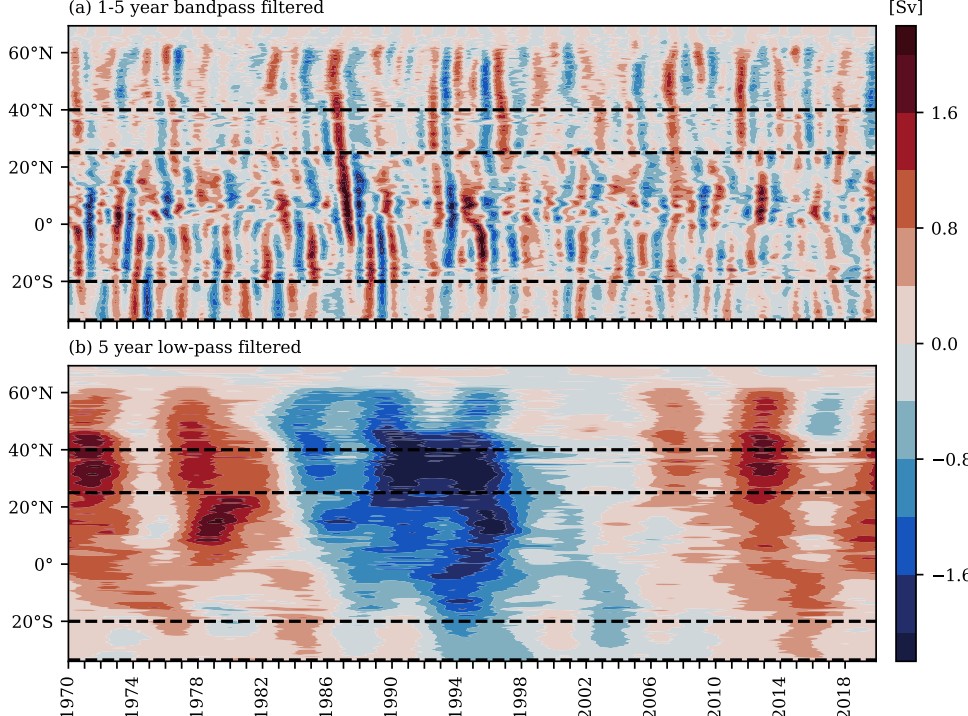

**Figure 2.** Hovmöller diagrams of the model-true upper NADW transport anomalies based on direct VIKING20X transports averaged over the 1000–3000 m depth range that have been filtered with (a) a 1–5-year band-pass filter, and (b) a 5-year low-pass filter (b). Horizontal dashed lines indicate the latitude interval considered in the following for both the North (25°N and 40°N) and South Atlantic (20°S and 33°S).

just changes the sign of the correlations. This is done in order to be more comparable to previous analyses which consider northward directed transports.

     The results in Fig. 4 show a strong correlation between the model transport anomalies and OBP that is confined to the steep and narrow continental slope region between 1000 m and 3000 m, matching the depth of the transport variations we are considering here. In addition, the correlations extend farther north than the 40°N limit of the considered model transports.

For both frequency bands, we find little correlation between NADW transport variations and OBP on the continental shelf itself. In the case of the 1–5-year band-pass filtered results, there are some modest negative correlations in the deeper ocean below 3000 m. Comparing the correlations from the two frequency bands indicates that seasonal variations are generally accessible as well since the lower limit of the considered frequency band does not affect the correlations at the continental slope. We additionally mark regions with a low statistical significance ($p$-value $> 0.05$) by hatching. This differentiation

further underpins the robustness and spatial coherence of the $p'_W$ signals along the continental slope associated with NADW





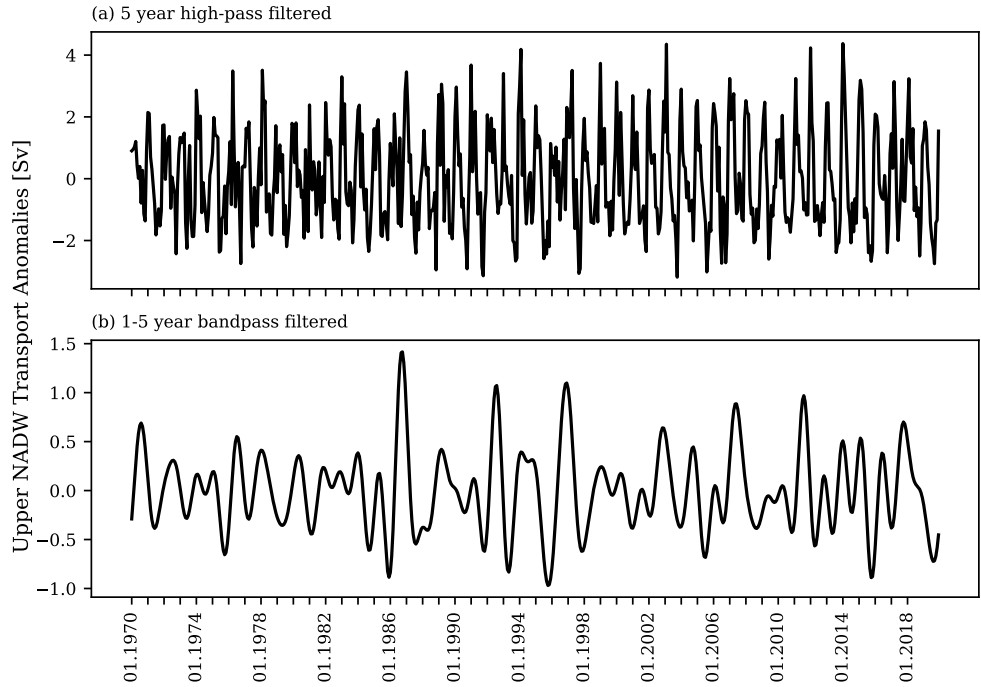

**Figure 3.** Time series of model-true monthly upper NADW transports in the 1000–3000 m depth range averaged between 25°N and 40°N from VIKING20X. Results are shown using a 5-year high-pass filter (a) or a 1 to 5-year band-pass filter (b).

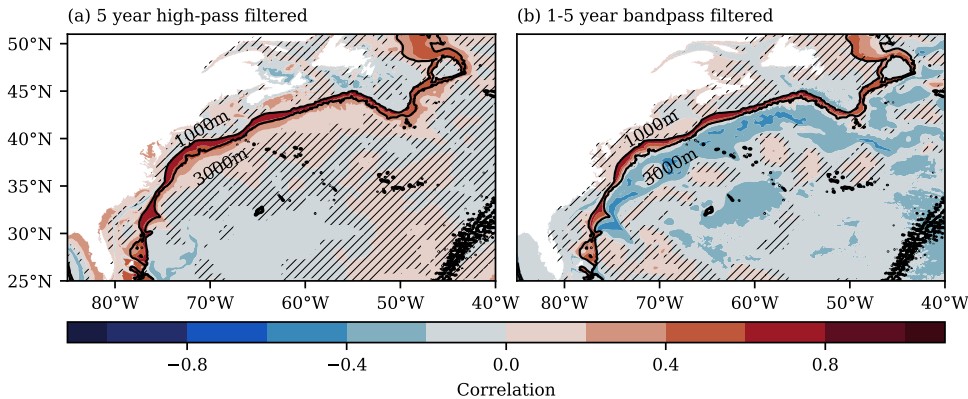

**Figure 4.** Pearson correlation between model-true upper NADW transport variations in the 1000–3000 m depth range averaged between 25°N and 40°N and OBP anomalies in the Northern Atlantic. Both NADW and OBP are either filtered using a 5-year high-pass filter (a) or a 1 to 5-year band-pass filter (b). Depth contours are indicated by black dotted lines and regions with a $p$-value over 0.05 are hatched. Note that we use the negative of the transport time series to be consistent with other analyses which usually consider northward directed transports.





transports. In general, results in Fig. 4 are consistent previous studies such as Roussenov et al. (2008) who, in contrast, assessed the northward-directed upper AMOC transports and thus find similar correlations.

Based on the above correlations, we can expect to be able to infer the NADW transport variations reasonably well from OBP anomalies along the slope. But since we are considering the total meridional transport between 1000 m and 3000 m,
based on Eq. (4), OBP anomalies at the western boundary need to be integrated vertically. Performing the integration using OBP anomalies from the western boundary, scaling with $(-f\rho_0)^{-1}$ and averaging over latitudes gives the time series shown in Fig. 5 (a) in dashed blue. Additionally, we show the direct model-based NADW transport anomalies in red for comparison. Note that for the sake of better visibility, we show a short time span from 1990 to 2019 only. Based on the results in Fig. 5 (a), we conclude that it is possible to reproduce NADW transport variations reasonably well (Pearson correlation: 0.76) with a
root-mean-square error (RMSE) of 1.13 Sv only compared to the full transport RMS of 1.62 Sv.

### 3.2 Regression Approach

Putting the analyses of the previous section into the context of satellite gravity measurements, however, a numerical integration as performed here based on Eq. 4 will not be feasible, since the smallest spatial scales are dominated by correlated errors. Even for future constellations that will consist of multiple satellite pairs such as MAGIC, the required spatial resolution is not
attainable. Instead, satellite measurements could possibly yield the average mass anomalies over the slope region. This means that deriving NADW transport anomalies will likely require a scaling procedure. One option might be to use model simulations to derive such a scaling relationship. To test the efficacy of such an approach, we average OBP anomalies in the slope region between 25° and 40°N, giving a single time series of OBP anomalies. Next, we fit a single scaling factor such that the OBP time series best reproduces the model NADW transport anomalies in terms of a simple linear regression. The resulting time
series is given in Fig. 5 (b). The scaling factor derived in this case is $-0.24$ Sv/Pa. Using this approach reduces the correlation between scaled OBP anomalies and transport variations only slightly by 0.02 and increases the RMSE by 0.34 Sv as shown in Tab. 1, thereby demonstrating that using the average OBP signal is a feasible approach.

So far, the analyses have focused exclusively on OBP signals along the continental slope. In principle, this has the advantage that background signal levels are smaller compared to the adjacent deep and continental shelf regions and thus also a larger
fraction of variability is related to changes in NADW. The disadvantage is that we focus on a target area with very limited cross-slope extent. As an alternative, signals on the continental shelf offer the possibility to improve the estimation of transport variability since the OBP variations on the shelf partly reflect the upper northward limb of the upper AMOC cell. As in principle upper and lower limb should compensate each other and hence share the same variations, adding information from the upper branch could in principle improve our estimate.

We therefore modify the regression approach by considering the OBP signals on the continental shelf together with signals at the slope. We do this by calculating averages over the two regions (0–1000 m and 1000–3000 m depth between 25°N and 40°N), multiplying each with its associated scaling factor and calculating the difference between the two. We use the difference, since the OBP signals on the shelf are related to transport variations in the opposite direction. The two scaling factors are determined again through a single regression to best fit the NADW transports. The resulting scale factors in this case are $s_{slope} = -0.13$

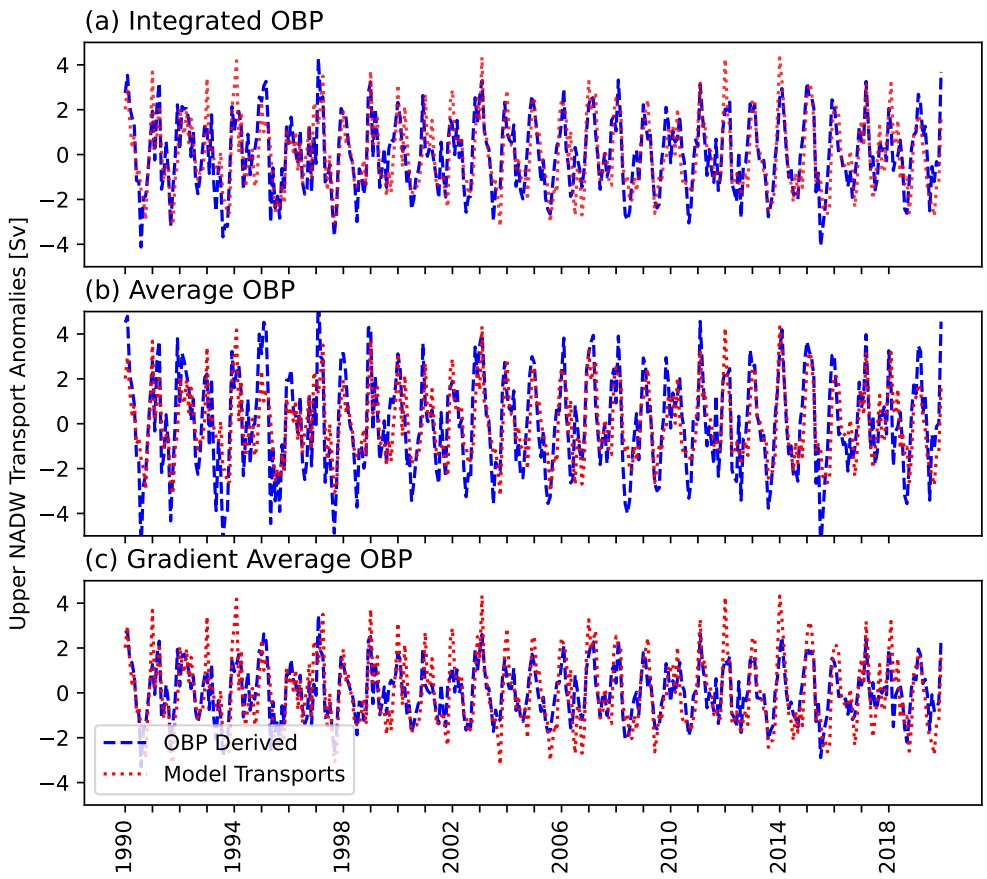

**Figure 5.** Upper NADW transport anomalies inferred through OBP anomalies from the western North Atlantic averaged between $25°$ and $40°$N (blue). (a) uses vertically integrated OBP anomalies along the continental slope. (b) considers the spatially averaged OBP anomalies from the slope region with a regression based scaling-factor. (c) uses the average OBP anomalies from the western continental slope and the shelf region with two regression-based scaling factors to best represent transport variations. Model-true transport anomalies are given in red.





**Table 1.** Summary statistics for comparisons of modelled and OBP-based transport anomalies. Tabulated are correlations and RMSE values between the model transport anomalies and OBP-inferred transports using either integrated OBP based on Eq. (4), average OBP from the continental slope through a regression, or a regression using both slope and shelf. Both North and South Atlantic are considered.

|  | Correlation | RMSE [Sv] |
|---|---|---|
| North Atlantic: |  |  |
| Integrated OBP | 0.76 | 1.13 |
| Regression slope | 0.74 | 1.47 |
| Regression slope & shelf | 0.75 | 1.07 |
| South Atlantic: |  |  |
| Integrated OBP | 0.63 | 1.10 |
| Regression slope | 0.64 | 1.67 |
| Regression slope & shelf | 0.72 | 0.97 |

Sv/Pa and $s_{shelf} = 4.2 \cdot 10^{-3}$ Sv/Pa. Note that the difference in scale between the two does not necessarily mean that there is little weight from the shelf region, since the variability on the continental shelf is of higher amplitude (see Fig. 1). Additionally, since we determine both factors in a single regression, $s_{slope}$ here deviates from the result in the previous section. The inferred upper NADW transport anomalies, calculated via

$$T'_{OBP} = \overline{p}'_{slope} \cdot s_{slope} - \overline{p}'_{shelf} \cdot s_{shelf} \tag{5}$$

are shown in Fig. 5 (c). Including the contribution from the shelf region increases the correlation with the model-based upper
NADW transport time series by 0.01 (to 0.75) but the RMSE between the two is reduced to 1.07 Sv (see Tab. 1). This result indicates that the larger shelf region can contribute to the estimation of upper NADW transport variations.

The suitability of the approach for application in satellite gravimetry remains to be tested. While it increases the spatial extent of the mass variations, it comes at the cost of a much higher noise level on the shelf (i.e., OBP signals that are unrelated to changes in NADW transports). In addition, the target region borders landmasses and thus increases potentially adverse
impacts of hydrological signal leakage and temporal aliasing—two well-known weaknesses of satellite gravimetry. As a compromise, one could still use the shelf signals but exclude the shallowest waters. Thus, investigating which approach and regional constraint is best suited for for satellite applications remains to be tested in end-to-end satellite simulation studies.

### 3.3 Deep Ocean OBP Signals

While the analyses of the previous sections indicate that satellite-based OBP measurements may allow to monitor variations
in upper NADW, Landerer et al. (2015) already presented an analysis based on monthly JPL-GRACE mascon solutions that specifically targeted the lower component of NADW (LNADW). Focusing on depths between 3000 and 5000 m at 26.5°N, the authors reported good correlation with RAPID-based transports, especially for the anomalously weak meridional transports





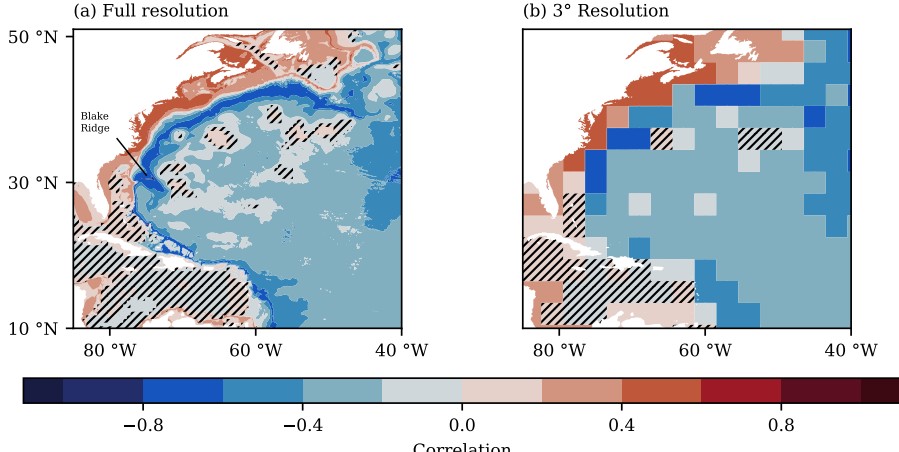

**Figure 6.** Correlation between model-true LNADW transport variations (3000–5000 m depth) averaged from 25–26°N and OBP. All time series are band-pass filtered to only include 1–5-year frequencies. (a) shows results using the full resolution from VIKING20X while (b) shows the results after reducing the resolution of OBP to 3° to be roughly comparable to current GRACE data. Hatched areas show regions with a $p$-value above 0.05.

during the boreal winter of 2009/2010. However, these results are not uncontested (Hughes et al., 2018), particularly given the effective resolution of the JPL-GRACE mascons (3°). Model-based analyses conducted over longer time-periods (Fig. 4, Roussenov et al. (2008); Hughes et al. (2018)) show a rather strict confinement of transport-related OBP signals to the continental slope region with overall smaller amplitudes than those analysed by Landerer et al. (2015).

Here, we test whether some of the identified correlations for LNADW and OBP in the deep-ocean can be reproduced using VIKING20X. First, we derive a time series for the LNADW transport variations. Since the results from Landerer et al. (2015) are based on 26.5°N, we average VIKING20X transports from 26° to 27°N and extract the transports between depths of 3000 and 5000 m. Next, we calculate the correlation to OBP for every grid point. The results are shown in Fig. 6 for the full resolution (a) and a grid with a reduced resolution of averaged 3° (b) to be comparable with the resolution of standard GRACE products.

For the high-resolution case, we find the expected negative correlations along the continental slope, although at somewhat greater depths than noted above. Most pronounced are the correlations around the Blake Ridge at about 30°N. For coarsened horizontal resolution, there are still some negative correlations around the same region since the spatial extent is sufficiently large. For most of the rest of the continental slope or the deeper ocean, correlations are either small or patchy. Based on these results, we conclude that strongly reduced LNADW transports (e.g., in winter 2009/2010), involving possible extensions of the slope signal to the wider Northwest Atlantic (McCarthy et al., 2020), may indeed be observable with satellite gravimetry. However, given the expected confinement of the $p_W$ signals to the slope region, in conjunctions with the usually small magnitude of about 1 hPa for a change of 1 Sv, using standard GRACE mascon solutions seems not suitable for reliable and regular monitoring of deep water transport variations. Nevertheless, it suggests that the signals we are interested in are not totally out





of reach and that data analysis approaches targeting precisely the elongated strip along the continental slope of North America
are worth pursuing.

## 4 South Atlantic

So far, all our analyses on the relation between OBP and meridional transport variations have been focused on the North
Atlantic, in particular on the region between 25°N and 40°N. In principle, however, deep water transports at other latitudes,
such as in the South Atlantic, can be inferred from boundary pressure signals as well. In this section we thus focus on upper
NADW transport anomalies in the South Atlantic and investigate how well they can be captured by OBP.

In particular, we consider NADW transports between the latitudes 20°S and 33°S. The southern limit of this range is de-
termined rather ad hoc by the geographic extent of the regional refinement in the VIKING20X simulation. Previous studies
indicate that the deep western boundary current breaks up into eddies at about 8°S (Dengler et al., 2004). Further south, NADW
is transported by propagating eddies which affect OBP variations and could thus negatively impact correlations. Reaching the
Vitória-Trinidade Ridge at about 20°S, the main part of the NADW flows further south as a reformed deep western boundary
current (Garzoli et al., 2015; Vilela-Silva et al., 2023). As a result, we chose 20°S as the northern limit in our investigations.

Similar to the analysis performed for the North Atlantic, we first derive a single time series of upper NADW transport
anomalies based on the model streamfunction by calculating the average transport between 33°S and 20°S. In contrast to
the previous section, however, we consider a depth range of 1100–3000 m since the transition between the northward and
southward directed limbs is about 100 m deeper in the South Atlantic, based on assessments of depth-profiles (not shown
here). For the same two frequency bands as before, the upper NADW transport anomalies are illustrated in Fig. 7. While the
amplitude is similar to the upper NADW time series derived for the North Atlantic, comparing Figs. 7 (a) and 3 (a) indicates
that the seasonal variations in VIKING20X are less dominant in the Southern Atlantic. As before, the time series shown in Fig.
7 (a) for the South Atlantic region is used in the following to assess the relation to OBP anomalies.

### 4.1 Integration Approach

We repeat the analyses from the previous Section 3.1 for the South Atlantic. As a first step, we calculate the correlation between
the time series of upper NADW transport variations and OBP at the western boundary. Fig. 8 shows the Pearson correlation for
all grid points with either a 5-year high-pass filter (a) or a 1–5-year band-pass filter (b) applied.

For both frequency bands, there is a moderate to strong negative correlation between the transport time series and OBP on the
continental slope. Note that the correlations are negative here, since we are in the southern hemisphere. Positive correlations are
in this case found on the continental shelf and are stronger compared to the northern Atlantic. Comparing the two frequency
bands, we find higher correlations using the 5-year high-pass filtered data which suggests that seasonal signals make up a
significant part of the connection between transports and $p'_W$.

As before, we reconstruct the upper NADW transport variations using integrated OBP anomalies on the western slope
following Eq. (4). The resulting time series is shown in Fig. 9 (a) together with the model-true upper NADW time series. The





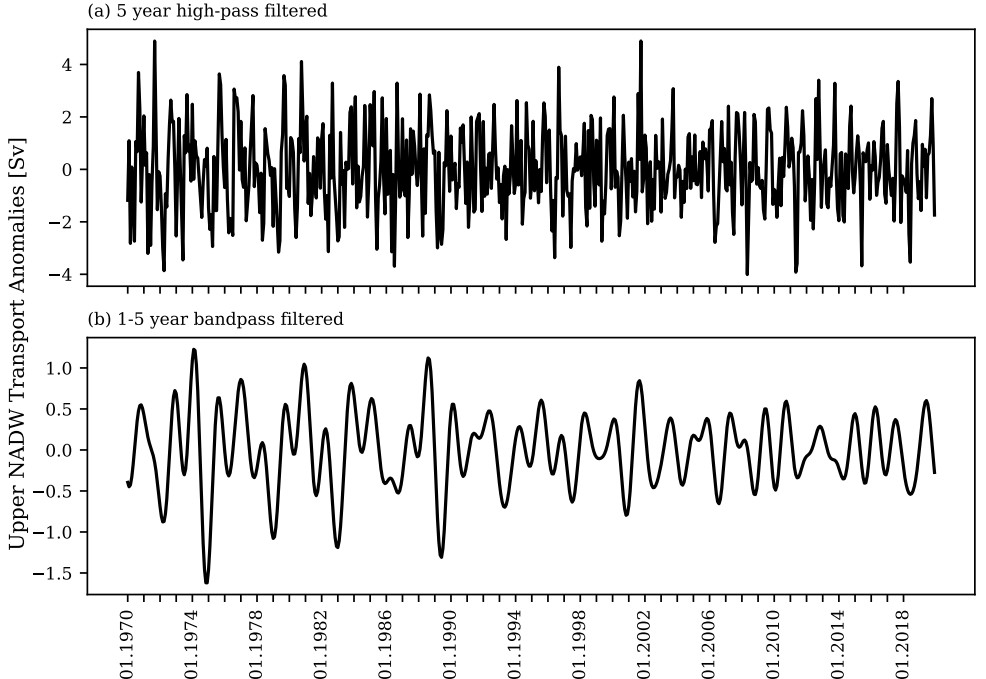

**Figure 7.** time series of model-true monthly upper NADW transport anomalies in the 1000 m to 3000 m depth range averaged between 20°S and 33°S from VIKING20X. Results are shown using a 5-year high-pass filter (a) or a 1 to 5-year band-pass filter (b).

OBP inferred time series shows a slightly weaker correlation of 0.63 compared to the one in the North Atlantic. Compared to the North Atlantic region, the RMSE of the OBP derived transport anomalies is similar with 1.10 Sv while the RMSE of
the model-true transport anomalies is smaller with 1.40 Sv. The OBP-derived time series thus underestimates the amplitude. Part of the reason for the difference in the statistics between the North and South Atlantic cases may be that the upper NADW transport variations have a less pronounced seasonality compared to the North Atlantic in Fig. 5 (a).

## 4.2  Regression Approach

These results change slightly when we consider the average OBP anomalies along the slope, which may be easier to sense with
the means of space gravimetry. Upon averaging OBP signals and determining a scale factor to best reproduce the upper NADW transport variations (0.72 Sv/Pa in this case), we obtain Fig. 9 (b). Although the correlation remains almost unchanged, the RMSE increases by 0.57 Sv as shown in Tab. 1. As evident, the amplitude of the inferred NADW transport variations is now rather overestimated.

Lastly, we also consider the contributions from the continental shelf. As shown in Fig. 8, this region is characterized by sig-
nificant correlations between upper NADW transports and OBP fluctuations. As a result, one can expect to see some improvements in recovered transports when incorporating signals on the shelf. As described in Section 3.2, we do this by calculating



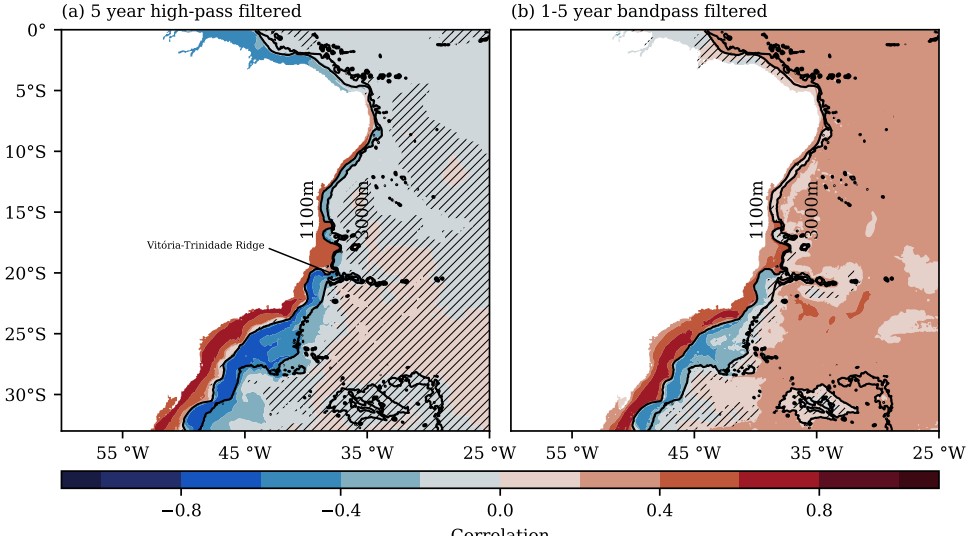

**Figure 8.** Pearson correlation between model-true upper NADW transport variations in the 1100–3000 m depth range averaged between 20°S and 33°S and OBP anomalies in the southern Atlantic. Both NADW transport anomalies and OBP are either filtered using a 5-year high-pass filter (a) or a 1 to 5-year band-pass filter (b). Depth contours are indicated by black dotted lines and regions with a $p$-value above 0.05 are hatched.

separate averages of OBP for the continental slope and shelf regions. We then compute two scaling factors through a single regression such that the OBP-based upper NADW time series best reproduces the true NADW signals. In this case, the scaling factors are $s_{slope} = 0.24$ Sv/Pa and $s_{shelf} = 0.15$ Sv/Pa and show, similar to the correlations in Fig. 8, that the shelf signal is given a significantly higher weight compared to the North Atlantic. The resulting transport time series is shown in Fig. 9 (c). Indeed, including OBP anomalies on the continental shelf improves both the correlation and the RMSE as indicated in Tab. 1 and seems to offer the best approach to infer the upper NADW transport anomalies through $p'_W$ anomalies in the considered region.

## 5 Synthetic time series of upper NADW-induced OBP variations

The analyses that we have presented so far suggest that inferring upper NADW transport variations through OBP anomalies along the western Atlantic is indeed feasible. However, it remains to be seen whether the variations can also be reliably tracked with future satellite gravimetry missions. Such tracking would involve several questions. These include the exact region to be considered as the target, the development of specific processing strategies to deal with the limited spatial resolution and possible spatial leakage, and the identification of a future mission concept that can indeed sense these OBP signals in the presence of other mass redistributions in the Earth system. Answering these points requires full end-to-end satellite simulations which offer direct control in terms of the target signal and the ability to consider adverse effects like sensor noise or aliasing artefacts

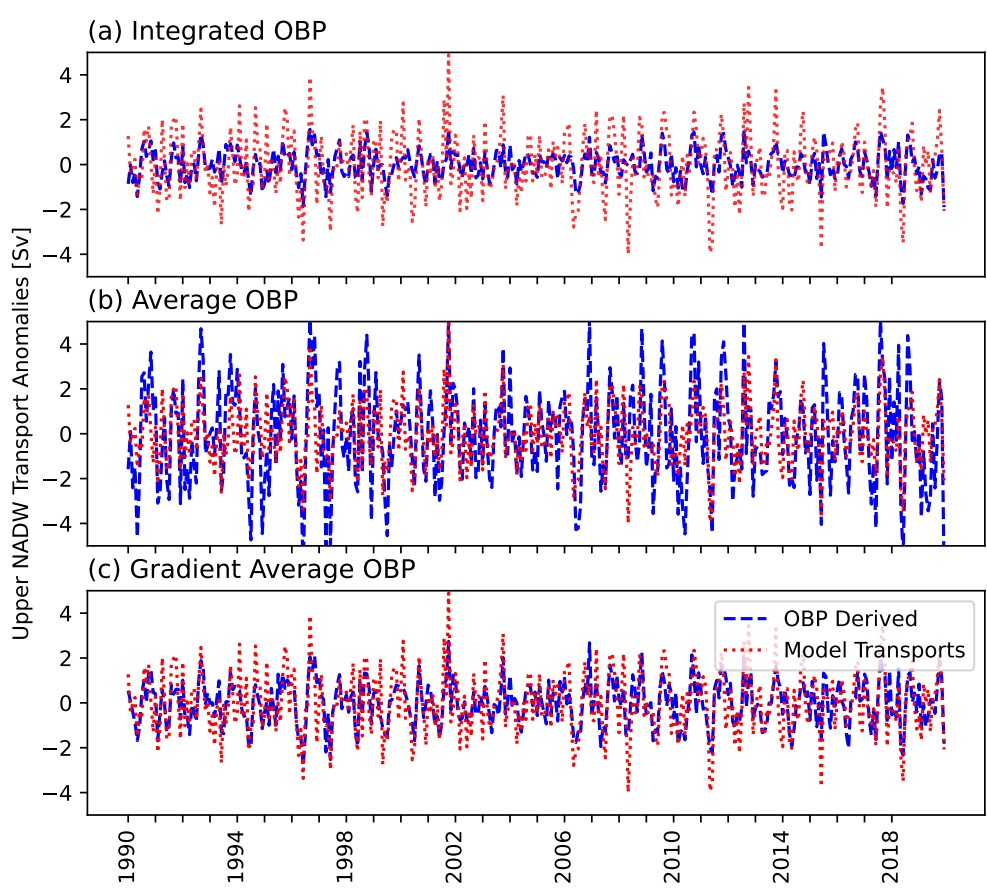

**Figure 9.** Upper NADW transport anomalies inferred through OBP anomalies from the western South Atlantic between 33° and 20°S (blue). (a) uses vertically integrated OBP anomalies along the continental slope. (b) considers the spatially averaged OBP anomalies from the slope region with a regression based scaling-factor. (c) uses the average OBP anomalies from the western continental slope and the shelf region with two regression-based scaling factors to best represent transport variations. Model-true transport anomalies are given in red.





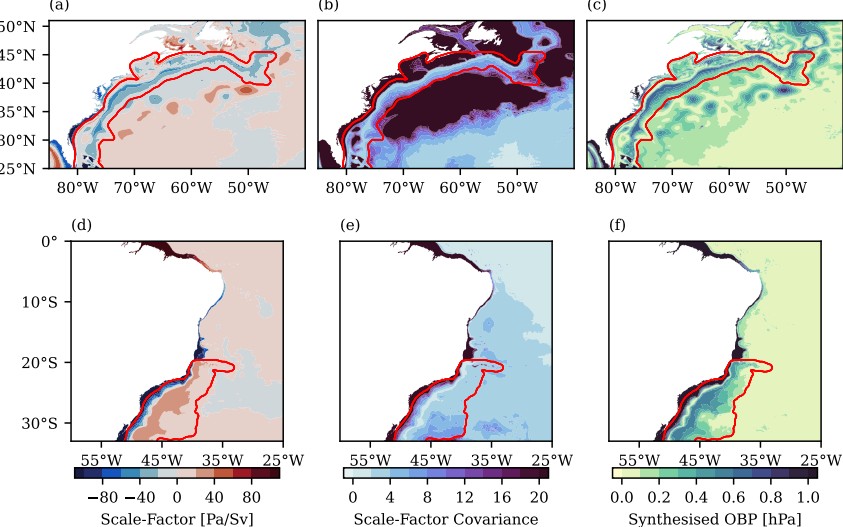

**Figure 10.** Results of the regression to derive a synthetic OBP time series that represents upper NADW transport variations in the North (top) and South (bottom) Atlantic. The scale factor from the regression of the NADW time series to OBP is given on the left and the associated uncertainty from the regression in the middle. The standard deviation of the synthetic OBP time series is given on the right. All subfigures include a possible selection of the region of interest as a red contour.

in a sequential manner. To facilitate such simulations, AMOC-induced OBP anomalies need to be prepared to be compatible with existing end-to-end satellite simulation setups. In this section we describe how such a preparation can be done based on simulated VIKING20X upper NADW transports and OBP anomalies.

The input to the simulations will be a time series of OBP anomaly grids for the two regions discussed above. These anomalies should contain only signals due to transport variations of upper NADW and no other non-transport related OBP signals. At the same time, the signal amplitudes should be as realistic as possible. Since the VIKING20X OBP anomalies also contain non-transport related signals, we create a synthetic OBP time series that meets both of these requirements. We do this by performing a regression of transport variations to OBP. In essence, this is the inverse approach to the regressions presented in

the previous sections. For both the North and South Atlantic regions, we take the single time series of model-true transport variations as shown in Figs. 3 and 7 (a) and perform a fit to the OBP time series at each grid point determining a scaling factor for each location. Next, we multiply the regression-based scaling factor at each grid point with the model-based transport time series to create a synthetic OBP time series. The resulting dataset has only the OBP variations due to upper NADW transport variations at each grid point and a realistic OBP amplitude. That is, the synthetic data combines the temporal behaviour of the

model transports with the correct signal strength of the OBP anomalies in the model but contains no other OBP signals.

The result of these computations for the North Atlantic are depicted in the top row of Fig. 10. The calculated scaling factor in Fig. 10 (a) essentially reproduces the results from Fig. 4. The uncertainty of the regression is shown in Fig. 10 (b). Smaller





values, which indicate a better fit and thus a more reliable result, are found mainly in the continental slope region. On the shelf and in the deeper ocean, the regression is not reliable. The standard deviation of the resulting synthetic OBP time series if given in Fig. 10 (c). This strip of variability along the slope is the target signal in satellite gravimetry simulation studies. In all subfigures, we include a possible selection of the region to be supplied to simulation studies as a red outline. This region is determined by selecting the area down to depths of 4000 m, including the slope region, and applying a Gaussian smoothing to create a single coherent area. As a result, the selected region contains the continental slope as the main target but also the signals on the continental shelf that may be of interest when considering the regression based on both regions. We base this selection on depth contours and not on the regression uncertainty as we want to include the shelf region in the end-to-end satellite simulations since they can improve the reproduction of transport variations.

Similarly, results for the South Atlantic are shown in Fig. 10 on the bottom row (d-f). Again, Figs. 10 (d), (e) and (f) show scale factor, uncertainty and standard deviation of the derived synthetic data, respectively. Notably, the target signal for satellite gravimetry seems to be slightly wider due to the smaller gradient of the continental slope. This is also reflected in the red outlined region in which the data is to be supplied to simulation studies. The region is determined in the same way as in the North Atlantic.

## 6 Conclusions

Deep water transports can be approximated through bottom pressure anomalies at the western boundary of an ocean basin, thereby providing an opportunity to monitor these transports through satellite gravimetry missions. That said, reliable estimates from this approach will likely remain elusive with standard GRACE / GRACE-FO monthly solutions, mainly due to the limited spatial resolution of the resulting time-variable gravity fields and derived bottom pressure anomalies. However, future gravity missions, especially those consisting of multiple satellite pairs such as planned in the MAGIC constellation, may allow monitoring NADW transport anomalies from space. One can expect trans-basin moorings (e.g., RAPID) to continue deliver highly accurate transport determinations, but the necessary arrays are costly and tend to be sparse, such that satellite-based observations may help to fill gaps in spatial coverage and provide continuous large-scale monitoring.

Work toward this objective will require dedicated end-to-end simulation studies to assess capabilities of the missions, develop processing strategies, or even refine requirements for future mission scenarios. We have here assessed the connection between western OBP and upper NADW transport anomalies in the VIKING20X model for the North and South Atlantic. Correlations with OBP in the region of the western continental slope confirm previous model-based studies in that the connection is strong mainly along the slope. While a theoretically precise estimation of NADW transport variations requires a spatial integration of OBP across the slope, almost identical results can also be achieved by using the average OBP anomaly from the slope region as it might be accessible through satellite measurements when applying an empirically derived scaling factor. Correlations between OBP-derived and direct model transports calculated in this way are 0.74 for the North Atlantic and 0.64 for the South Atlantic. Including average signals from the continental shelf region can improve results for both the North and South Atlantic. Although the improvement in correlation for the north is only modest (0.01) it is more significant in the south where the





correlation increases by 0.08. This would suggest that inferring upper NADW anomalies through OBP is not only feasible for the North Atlantic, as it is often considered, but also promising in the South Atlantic.

Based on the results using the VIKING20X data, we have also estimated a synthetic time series of OBP anomalies that can be used in satellite simulation studies. To that end, we have performed a regression of modelled upper NADW transport anomalies to OBP at each grid point. This way, the resulting synthetic OBP data includes only the variations due to transport variations while having realistic OBP amplitudes. In addition, we have delineated a spatial mask for the region of interest based on depth contours, such that we can supply the data in the target region only.

Although our results suggest that the connection between OBP and deep water transport anomalies in VIKING20X is robust, there are components that are not captured by this analysis. For one, we have focused on western boundary pressures, as they contain most of the transport-related OBP signals. Neglecting eastern signals introduces small errors in the transport determinations and precludes the removal of pressure signals associated with basin-wide modes or ocean mass changes Bingham and Hughes (2008). As a result, estimates of $p_W$ may include signals not related to overturning. Since the synthetic OBP data to be provided for satellite simulation studies only contain overturning-related signals, this additional complication would not be captured in the simulation studies. Nonetheless, we believe that feasibility tests in satellite simulations should initially start with a clearly defined target signal, leaving the treatment of somewhat secondary aspects for later stages of refinement.

We have also not considered trend signals in meridional transports in view of the nominal satellite mission lifetime of just 5 years. Assessing whether trends are accessible through satellite measurements or sensitivity estimates is therefore not possible. While this question may become more relevant in the future, estimating subtle trend signals from satellite gravimetry is generally a matter of delicacy due to superimposed signals from, e.g., regionally variable barystatic sea-level rise, and glacial isostatic adjustment of the solid Earth (Chen et al., 2022).

Lastly, connecting average OBP from the western continental slope to NADW transport variations required an empirically derived scaling based on the actual model transports. While such an approach is of course feasible for measurements in simulations, deriving such a scaling applicable to actual measurements will certainly require more careful assessment. Despite these challenges, the work presented here is deemed an important first step for simulation studies, offering a foundation for the thorough assessment of future satellite gravimetry missions in monitoring deep water transports and AMOC variability, which can now be included in synthetic time-variable gravity field data-sets as the ESA Earth System Model (Dobslaw et al., 2015) which will be the basis for the upcoming mission performance evaluation studies for the MAGIC constellation.

*Data availability.* The data to reproduce the presented results and figures is available via Shihora et al. (2024). This also includes a gridded version of the synthetic OBP data for the North and South Atlantic. For more detailed VIKING20X information we refer the reader to Biastoch et al. (2021).



*Author contributions.* Main text and analysis performed by LS with contributions from all co-authors. Simulations are provided by GEO-MAR. HD and MS conceptualized the work, TM, ACH and RH gave valuable input to the methodology. All co-authors contributed to review and editing.

*Competing interests.* There are no competing interests.

390  *Acknowledgements.* This work has been supported by the German Research Foundation (Grant No. DO 1311/4-2) as part of the research group NEROGRAV (FOR 2736).



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
