# Peer review of "Relating North Atlantic Deep Water transport to ocean bottom pressure variations as a target for satellite gravimetry missions"

_EGUsphere, 2024_

## Author Response (AR1)

**Reviewer 1**

**Major Comments**

1. The authors refer throughout to North Atlantic Deep Water (NADW) transport as the quantity being measured/estimated. Although this makes for a neat narrative, speaking as an oceanographer, I think it relies on a simplistic, cartoonish view of the ocean's circulation, one that perhaps applies to some extent to the time-mean flow, but certainly not the time-variable component. The ocean's actual circulation and the anomalous zonally-integrated meridional flows over a particular depth range at a particular time, while perhaps incorporating some fluctuating NADW signal, are likely composed of a much messier mixture of water masses. And, indeed, the net southward flow probably results from an imbalance of northward and southward flows of different water masses, some not readily classified. In my view, it would be more accurate and scientifically correct to remain agnostic as to the water masses involved, and refer simply to the zonally-integrated meridional transport, in the title and elsewhere. The fact that this range likely includes some fraction of NADW, and the fact that meridional transport fluctuations may, to some undetermined extent, reflect more or less NADW flowing south, could be discussed in the introduction. There is a link but a highly qualified and uncertain one without a lot of further work.

*Thank you for the suggestion. Our main goal in referring to NADW was to have a well known and concise term to refer to. However, we agree that a more general description of the variations in the meridional transports should be used to avoid confusion. Instead we have adapted the manuscript and now refer simply to MVT which is inspired by the term MHT for 'meridional heat transport'. We only mention NADW in the general description of the overturning circulation and mention that NADW is likely a significant contribution to the zonally integrated transports, albeit not the only one.*
*We believe that this strikes a good balance between generality in referring to the involved water masses without being too cumbersome in the text.*

2. My second, "major" comment concerns the second aim of the work: The authors do not make clear why an extracted AMOC-related OPB signal is required for the end-to-end simulations. Why not just use an ocean model that naturally includes this signal along with other OPB variability? Although I think I know the reason, that motivation needs to be made clear for this section to make sense to the wider readership.

*Thank you for the comment. Indeed, the choice of separating the transport-related signals from the rest of the OBP variations in the ocean model might seem over complicated. We agree that this could be better motivated in the text itself and have added our reasoning to the beginning of Sect. 5. The reason we propose this approach lies mainly in the fact that this offers more control over individual noise contributions in the satellite simulation studies. In order to develop e.g. specialised retrieval approaches in the simulations, it is useful to have a clearly defined target signal which has no other noise contributions as a first test-case. Additional noise sources can then be added to the simulation setup in subsequent steps. Using an ocean model which includes all OBP signal contributions would complicate the interpretation of the simulation studies significantly.*

**Minor comments**

L20: Incorrectly formed citation.

*Thank you for the correction. The error has been fixed in the manuscript.*

L20: "contributed" -> "contributes".

*Corrected.*

L28: It would be good to specifically introduce the RAPID array here as it is the most comprehensive of the observational systems.

*We have added a specific reference to the RAPID array on l. 30.*

L75: Rather than stating "quite well" it would be good to quantify the agreement found by BH08.

*Agreed, the value from the reference has been included.*

L76: It would be worth emphasising that we can only recover anomalies by this method, and mention the prime notation.

*Thank you for the suggestion. In line with the corresponding comment from Reviewer 2, we have updated the description of the notation.*

L81: Do you mean the Deep Western Boundary Current in this depth range?

*Yes, this has been updated.*

L86: The strong relationship with US east coast sea level, as first demonstrated by Bingham and Hughes (2009) and subsequently supported by many studies, also lend credence to the dominance of the WB pressure.

L95: Bingham and Hughes (2008) also showed a strong correspondence between the upper and lower layer transports.

*Both of the above mentioned references have been updated in the text.*

L114-115: Place commas after "computed" and "component".

*Corrected.*

L115-116: Description of OBP calculation could be clearer. It is not entirely clear what is meant by 2-D surface density.

*Thank you for the suggestion. The description is updated to hopefully avoid confusion.*

L117: "filtered" -> "filtering".

*Corrected.*

L121-2: Sentence starting "Overall" is unclear/awkwardly phrased.

L147-9: The sentence starting "Although" is too long and difficult to parse. The z vs. sigma issue is rather a distraction. I would delete or make it clear by rephrasing that Biastoch is an example of the latter (if that's the case).

*We have updated the description of the previous analyses using VIKING20X in both cases.*

Figure 4 caption, last sentence: "with previous analyses" sounds better.

*Corrected.*

L185: "1.13 Sv only" -> "only 1.13 Sv". Is the "only" really justified here given it is quite close to the RMS itself? % of variance accounted for (skill) might be a useful additional (better) metric to provide.

*This is in line with the main comment from Reviewer 2. We have, for one, changed the text to reflect the fact that there are significant noise contributions and have additionally included the PVE value as well as the RMS values in Tab. 1.*

L190: With regard to the statement beginning "Instead" it would be useful here to support this statement with the results shown in Figure 9 of Pail et al (2015) and Figure 13 of Daras et al (2024). If not here, then I think the results shown in these two Figures deserve a more detailed discussion/ examination in the introduction, being the only two studies (to my knowledge) that have examined how the slope bottom pressure is impacted by spherical harmonic truncation and noise. It is not as though examinations of these two are not included because there is a lot of additional/better relevant literature to be considered.

*Thank you for the suggestion, we have added references to both publications and the result in terms of future mission capabilities on what is now l. 198.*

L196: Remove the 0.02 value, or just give the correlation itslef.

*Corrected.*

L210: It is worth noting that when expressed in Sv / cm the shelf scale factor -0.42 is similar to the -0.59 scale factor for the relationship between coastal sea level and the AMOC at 42N found by Bingham and Hughes (2009), with the higher magnitude value found in that study plausibly explained by the larger amplitude of coastal sea level compared with bottom pressure averaged over the shelf. Note, that paper (Figure 1a) also shows a very clear OPB signal along the slope between 1300 and 3000 m.

*Thank you for the suggestion, we have noted the similarity to the results presented in BH09.*

L210: In BH08, we removed the depth average slope pressure before computing the meridional transport in the upper and lower layers. To an extent, the gradient method achieves the same thing - removing the common signal, although not as effectively as computing the slope average pressure directly (which, or course, is not possible with GRACE like data). The reduced amplitude of the OPB recovered transport (5c) and the lower RMSE of the gradient approach compared to the average OPB transport (5b) seems to support that. I wonder if the gradient approach could be further improved by first removing the average signal across the shelf and slope.

*Thank you for the suggestion. This might explain the contribution from the shelf region. Especially since the correlations in that region are rather small, some improvement may come from the common signal.*

L224: "may allow to monitor variations" -> "may allow variations to be monitored"

*Corrected.*

L225: Delete "already".

*Corrected.*

L233: Would be good to show this time series.

*Agreed. We have updated Fig. 6 to include the time series. Since we show the time series for the other analyses it should be included here as well.*

L238: "as noted above" - please be more specific.

*Corrected.*

L294: Perhaps it is beyond the scope of work, but it would be interesting to consider why the scale factors are so different for the SA.

*The differences between the North and South Atlantic can also be found in the correlation patterns of Figs. 4 & 8. Especially the correlation in the shelf region. Some of the difference may be attributed to the somewhat different overturning structure in the south and the less pronounced seasonal signal.*
*Since it is not very relevant for satellite simulations, we have not further considered the differences here but they may warrant additional studies nonetheless.*

L300: "suggest" - "confirm previous analyses" is perhaps more accurate.

*Corrected.*

L318-9: While I agree that the dataset will primarily contain transport related to OPB, and as such will be fit for the proposed purpose, it is perhaps an overstatement to say that the resulting dataset *only* has OBP variations due to the upper NADW (zonally-integrated meridional) transport, since it may also contain OPB variations that are correlated with the transport but are not dynamically/ geostrophically related to the transport variations (eg balanced, basin-wide OPB patterns and other wind-driven signals).

*Thank you for the correction. While a significant contribution of the signal in the synthetic OBP data will be due to overturning, we agree that the statement made in the text is too strict. The text has been updated to include the additional signal sources which might still be present in the data set.*

L345: Perhaps "provide continuous large-scale monitoring" should be qualified with the appendage "should satellite gravimetry become operational" or similar.

*Yes, an important clarification that has been added.*

L349: "previous model-based studies" - cite them.

*Corrected.*

L352: "as it" -> ", which"; "when" -> ", by".

*Corrected.*

L360: "only the variations due to transport variations" - as before, strictly speaking this is not necessarily true - there could be other correlated factors compensating or driving OBP.

*Corrected.*

L362: Not clear what is meant by "we can supply the data".

*We have updated the phrase to hopefully avoid confusion.*

L365: "precludes the removal of pressure signals associated with basin-wide modes" - as mentioned above, the gradient approach does this to some extent which may be the reason for the improved agreement.

*Agreed. While some part of the common signal is likely removed, a (here undetermined) contribution will remain and not be accessible. This should now be clarified in the text.*

L366-7: Incorrectly formatted citation.

*Corrected.*

L379: "is deemed an important first step" - in light of Pail et al (2015) and Daras et al (2024) it is perhaps a little inaccurate to say the work is a "first step". It is an important contribution; though I always feel it is better to let the reader judge the importance of one's work, rather than it be the authors doing the deeming! In fact, since the final sentence is also too long, I would recommend a more elegant ending comprised of two shorter sentences.

*Thank you for the correction. Indeed, 'an important first step towards full end-to-end satellite simulations' may be more appropriate. However, since the last few sentences were a bit cumbersome, we have decided to rephrase that part to be less suggestive.*

**Reviewer 2**

**Major Comments**

From the results shown and discussed, I would think that inferring overturning transports from western boundary OBP variability can leave a substantial part of the transport variability unaccounted for. However, that is not the tone of the paper and the opposite message permeates the presentation and interpretation of results. A few concrete examples serve to substantiate my view.

183-185 Based on RMSE of 1.13 Sv and RMS of 1.62 Sv for total transport, my reading of these results is that large errors will be involved in inferring transports from OBP anomalies, i.e., opposite to what is conveyed in this sentence. Results would probably be even worse if not averaged over 15 degrees of latitude (25N-40N) as shown in Fig 5a. Furthermore, most of the skill is probably associated with the annual cycle. I am curious about what the results would show if a mean annual cycle was removed from the analyzed time series.

195-197 The regression method in fig 5b is even worse with the RMSE of 1.47 Sv. Results do get a little better for the regression using slope and shelf (fig 5c). Nevertheless, all cases seem to show signal-to-noise ratios of around 1.5 or less. To be able to assess the value of OBP, it would be good to include RMS of respective real transport and OBP-based results in Table 1, for direct comparison with RMSE values. An alternative would be to show the ratios of RMS to RMSE, as a measure of signal-to-noise ratio for each method.

The above examples deal with North Atlantic results but the discussion of South Atlantic results also needs to be clarified along the same lines.

300-301, 356-357 These sentences are ambiguous and, unless they can be made more quantitative, hardly justified. The text needs to quantify, at the very least, what level of uncertainty is associated with such OBP-based transport estimates. Providing numbers on how much of the variance can be explained by the various approximate OBP-based methods could set the discussion on firmer grounds.

*Thank you for the thorough assessment of the manuscript. We agree that the analyses as well as the proposed synthetic dataset present an optimistic estimation of the connection between bottom pressure and transport variations. This is done with some intention.*

*Several complicating aspects are not accounted for in the work that we present. Some of them are mentioned explicitly in the beginning, e.g. neglecting eastern boundary pressure variations or some modes of variability. Some are only implicit, such as including the seasonal signal in the analysis which, as rightly pointed out, is a significant part of the signal amplitude and thus strengthens the connection between transport and bottom pressure. In addition, there are some other aspects which were not explicitly mentioned such as additional complications due to inter-annual changes in the amplitude of the M2 tide in the region of the western slope region (Schindelegger et al. 2022 10.1029/2022GL101671).*

*Our reason for doing so lies with the goal of using the bottom pressure signatures in the preparation of future satellite gravimetry missions and their simulations. Even with this positivistic view of our work, it is not at all clear how well future missions or constellations such as MAGIC might resolve OBP in such a confined region. Thus, neglecting additional complications and*

*including the seasonal signal should be considered as a first step to test new processing strategies. They should consequently be seen more as an upper bound if you will.*

*While we are aware that the reality will be much more complicated, we agree that this may not be sufficiently clear in the text. To avoid confusion in that matter and be more transparent, we propose to adapt the manuscript in several instances.*

*The text in subsections 3.1 and 3.2 has been updated to better reflect the challenges that remain when estimating meridional transport variations from OBP. Additionally, we have included the RMS values for the relevant time-series in Tab. 1. Lastly, we have updated the discussion in the last section to better reflect the impact that the seasonal signal and noise have.*

**Minor Comments**

6 "gravimetry"

7-8 Not clear what you are assessing: the connection between North and South Atlantic OBP or the connection of OBP to NADW transport?

9 "spatially averaged" over what regions, depths? Specify in the abstract.

16 "comprises a fine-structured"

19-20 "Southern Ocean (Buckley and Marshall, 2016)"; "which contributes"

34 Delete "already"

44 Explicitly state what are "these OBP variations".

56 Here and elsewhere, "northern Atlantic" should be "North Atlantic".

61-63 Sentence is hard to follow and could be rephrased for clarity.

72 "Coriolis parameter that is induced"? Awkward phrasing.

*Thank you for the corrections. All comments from above have been implemented in the text.*

Equation (3) Define prime variables. It is also never explained in the paper how the transports are calculated when there are two "western" boundaries at 1000-3000 m depths, which is the case at latitudes around 44N (see, e.g., fig 4)?

*Thank you for the comment. We have updated the description in Sec. 2.1 to introduce the prime-notation.*
*Regarding the analysis in the presence of two 'western boundaries': we do not go into details regarding the case as it lies outside of the region we consider when estimating the transports based on bottom pressure anomalies. The northern limit we use is 40°N which excludes the complication at around 44°N as well as others at e.g. 48°N.*
*But we do agree that this might warrant a more careful assessment when considering what we call in the text the 'integration approach'. For a regression based analysis however, which is based on spatially averaged OBP anomalies, dealing with such a 'double boundary' is more straight forward as both regions are simply part of the averaging. Since that is the basis of the synthetic dataset we*

*aim to provide, this should not pose a problem, even if our analyses were to extend further north.*

85-86 Clarify the text on what is meant by "as well as, e.g., basin wide modes".

*We changed the phrasing of the remark and added an exemplary reference to the study of Stepanov & Hughes (2006, 10.1029/2005JC003450) who have investigated a mode of variability that is relevant in this context.*

117 "Furthermore,..."

143 Which current? Clarify the text.

Figure 2  Clarify somewhere in the text that 1-5 year band-pass results do not contain contributions from the annual cycle?

176 "consistent with previous"

*Thank you for the corrections. All comments from above have been implemented in the text.*

207-208 I don't follow these explanations. I thought the regression would yield appropriate signs for the scaling factors.

*That is correct. In setting up the equation for the regression it is not relevant whether the sum or difference is used since it would simply change the sign of the regression coefficient. We choose the difference here because we expect variations in bottom pressure connected to the southward directed transport on the slope and shelf to be of opposite sign. Setting up the regression equation in this way might thus be more intuitive. In terms of the final result, this should not matter.*

243 "conjunction "

279-280 I think the second reference to "RMSE" should be "RMS"? But then the final sentence on line 280 does not follow, so perhaps both references to "RMSE" should be "RMS".

Figures 5 and 9. State in the caption that you are analyzing 5-1 year band-pass series?

*Thank you for the corrections. All comments from above have been implemented in the text.*

311 I don't know what that means?

*Our goal here is to provide a 'clean' target signal to be used in simulations that does not contain OBP variations which might be considered 'noise' in this context. In other words, the input to the satellite simulations should represent only OBP variations that arise through variations in meridional transports.*

*As mentioned above, we do this to start with the simplest (or one might say optimistic) case in the following simulation studies.*

*We have rephrased the section in the manuscript to explain our intentions better.*

332-332 These details should be provided solely in the figure captions. This comment applies to many other instances in the text where redundant information is provided, detracting from the text main story.

*Thank you for the suggestion. We have updated and shortened the reference to Fig. 10. Additionally, we have simplified other instances in the hope that the text becomes more streamlined.*

343 "continue delivering"

366-367 Fix reference format.

377 "measurements in simulations"?

378-384 Please fix run-on sentence.

*Thank you for the corrections. All comments from above have been implemented in the text.*